# The Swedish Standardized Course of Care—Diagnostic Efficacy in Esophageal and Gastric Cancer

**DOI:** 10.3390/diagnostics13233577

**Published:** 2023-12-01

**Authors:** Philip Kanold, Nils Nyhlin, Eva Szabo, Michiel van Nieuwenhoven

**Affiliations:** 1Department of Internal Medicine, Division of Gastroenterology, Örebro University Hospital, SE 70116 Örebro, Sweden; 2Department of Gastroenterology, Faculty of Medicine and Health, Örebro University, SE 70182 Örebro, Sweden; michiel.van-nieuwenhoven@regionorebrolan.se; 3Department of Surgery, Faculty of Medicine and Health, Örebro University, SE 70182 Örebro, Sweden; eva.szabo@regionorebrolan.se; 4University Health Care Research Center, Faculty of Medicine and Health, Örebro University, SE 70182 Örebro, Sweden

**Keywords:** fast track, standardized course of care, alarm symptoms, esophageal and gastric cancer, positive predictive value

## Abstract

Fast-track pathways for diagnosing esophageal or gastric cancer (EGC) have been implemented in several European countries. In Sweden, symptoms such as dysphagia, early satiety, and other alarm symptoms call for a referral for gastroscopy, according to the Swedish Standardized Course of Care (SCC). The aim of this study was to evaluate the diagnostic yield of the SCC criteria for EGC, to review all known EGC cases in Region Örebro County between March 2017 and February 2021, and to compare referral indication(s), waiting times, and tumor stage. In our material, EGC was found in 6.2% of the SCC referrals. Esophageal dysphagia had a positive predictive value (PPV) of 5.6%. The criterion with the highest PPV for EGC was suspicious radiological findings, with a PPV of 24.5%. A total of 139 EGCs were diagnosed, 99 (71%) through other pathways than via the SCC. Waiting times were approximately 14 days longer for patients evaluated via non-SCC pathways. There was no statistically significant association between referral pathway and primary tumor characteristics. The results show that a majority of the current SCC criteria are poor predictors of EGC, and some alarm symptoms lack a sufficiently specific definition, e.g., dysphagia. Referral through this fast track does not seem to have a positive impact on disease outcomes.

## 1. Introduction

Upper gastrointestinal (GI) cancer survival remains relatively poor despite both diagnostic and therapeutic advancements [1,2,3]. Prognosis is predominantly determined by tumor stage at diagnosis, and, unfortunately, a large proportion of upper GI cancers are diagnosed at advanced stages where treatment options are limited [4,5,6]. Specifically, esophageal and gastric cancer (EGC) represent the fifth and seventh most common malignancies worldwide, respectively, with disproportionately high mortality rates [7]. Among incentives to improve disease outcomes, several countries have implemented fast-track referral pathways for patients with alarm symptoms [8,9,10]. However, it is still unclear which symptoms are accurate predictors of malignancy and whether this approach is clinically meaningful.

In 2016, Sweden implemented the Standardized Course of Care for EGC (SCC-EGC), a fast-track pathway aimed primarily at reducing delays in the diagnostic and therapeutic process [11]. Current guidelines prompt immediate referral to gastroscopy in the presence of upper GI alarm symptoms; recent-onset dysphagia must be evaluated by gastroscopy, whereas emesis or early satiety > 3 weeks, severe unintentional weight loss, gastrointestinal bleeding, iron deficiency anemia or radiological findings suggestive of esophageal or gastric cancer should be evaluated by gastroscopy. A referral to a surgical department should be done when a well-founded suspicion is present based on either clinical or histopathological results.

The potential benefits of a fast track, with respect to diagnostic delay and disease outcome, were evaluated in 2020 by the Swedish Agency for Health Technology Assessment and Assessment of Social Services. It was concluded that although a fast track appears to reduce diagnostic delay, there is no evidence that disease outcome is simultaneously improved [12]. Other studies have been conducted in other countries with similar findings [8,13,14].

Early cancer detection, which is considered paramount to improving disease outcomes, is clinically challenging due to several reasons [15]. For instance, EGC is often asymptomatic during the early course of disease [16,17]. Furthermore, the initial presentation may be highly unspecific [18]. These factors contribute to delays in the diagnostic process. In addition, patient delay is considered one of the most critical prognostic factors in EGC and is difficult to target [19].

It has previously been demonstrated that endoscopic evaluation of upper GI symptoms is frequently performed with inappropriate indications [20,21]. Furthermore, even in patients with apparent ‘alarm’ symptoms suggestive of EGC, actual cancer yield tends to be low (<5%) [18,22,23]. This imposes a significant strain on healthcare resources. Thus, uncertainty exists regarding the diagnostic yield and effectiveness of the recently implemented SCC-EGC. It is also uncertain how well its entry criteria (alarm symptoms) predict malignancy. Furthermore, it is unknown whether esophageal and gastric cancers are primarily diagnosed through the SCC or via other routes. Finally, the differences between fast track and other routes are also unknown with respect to the diagnostic interval, symptoms, and disease outcome. Increased knowledge of the predictive values of SCC-EGC entry criteria may help clinicians prioritize referrals, thus using limited healthcare resources more wisely. Moreover, a better understanding of diagnostic pathways for EGC in clinical practice may be used to revise the current criteria, which may lead to an improvement in the fast-track referral pathway.

The primary aim of this study was to evaluate esophageal and gastric cancer yield in a population presenting with alarm symptoms who were referred for urgent gastroscopy according to the SCC-EGC guidelines, as well as to determine the positive predictive values of the SCC criteria. Second, to compare esophageal and gastric malignancies found via the SCC pathway with those found via other, non-fast-track pathways, with respect to referral indications, diagnostic and patient delay, and tumor stage.

## 2. Materials and Methods

### 2.1. Study Design

This was a retrospective review of medical records of all SCC-EGC referrals and endoscopy outcomes from the three endoscopy units in Region Örebro County (RÖC) from 1 March 2017 to 28 February 2021. Data were also collected from a database with all known EGC cases in RÖC during the study period.

Referrals came from both general practitioners and hospital-based specialists. All referrals within RÖC were reviewed centrally at the Örebro University Hospital by experienced gastroenterologists.

Gastroscopies were performed by gastroenterologists or surgeons working at the endoscopy units in RÖC.

### 2.2. Data Collection

Only patients living in RÖC were included in the study. For each patient, reasons for referral, waiting times, and background data, including laboratory results, were manually extracted from medical records. Gastroscopy outcomes were collected, and pathology reports were reviewed.

Anemia was defined as the most recent hemoglobin (Hb) value of <120 g/L for females and <130 g/L for males within the last month of the referral date. GI bleeding was defined as the presence of melena, hematochezia, hematemesis, or positive fecal hemoglobin (f-Hb). Dysphagia was based on the description in the referral and separated into esophageal dysphagia, oropharyngeal dysphagia, and dysphagia of uncertain location. Although the SCC criteria explicitly defines weight loss as having to be “severe” and vomiting or early satiety lasting a minimum of three weeks, these precise definitions were often impossible to extract from the referral. Therefore, the description of any symptom was included.

Risk factors such as current or past smoking or alcohol abuse were collected based on either the referral text, the patient’s record of diagnoses, or baseline information in the medical records. Obesity was defined as a body mass index (BMI) of ≥30 kg/m^2^ and/or a diagnosis of obesity at any point in time. Risk factors were assumed to be absent if they could not be extracted from any of these sources.

Tumors were classified according to the TNM system version seven until 2019 and version eight thereafter. The first known TNM stage from the electronic records was used for the data collection.

For the comparison of differences in delay between SCC and non-SCC referrals, the diagnostic interval (DI) was defined as the interval from the date of the initial referral (for any diagnostic procedure) till cancer diagnosis.

### 2.3. Statistical Analysis

Statistical analyses were performed using SPSS (IBM Corp. Released 2020. IBM SPSS Statistics for Windows, Version 27.0. IBM Corp: Armonk, NY, USA). The positive predictive values (PPVs) of SCC-EGC entry criteria were determined by dividing the number of symptomatic patients with EGC with all symptomatic patients. Differences in the distribution of categorical variables were determined using Pearson’s Chi-squared test or Fisher’s exact test if any parameter had an expected frequency < 5. Continuous variables were tested for normal distribution using the Shapiro–Wilk test. An unpaired *t*-test was used to analyze normally distributed data, and the Mann–Whitney U-test if the data were not normally distributed. A *p*-value of <0.05 was considered statistically significant for all tests.

## 3. Results

### 3.1. Patient Inclusion and Characteristics

A total of 946 gastroscopy referrals, according to the SCC-EGC, were identified between 1 March 2017 and 28 February 2021. After the exclusion of non-eligible referrals (see Figure 1), 856 unique gastroscopy referrals remained for analysis.

Fifty-three patients had esophageal or gastric cancer, resulting in a 6.2% EGC yield within the SCC-EGC referrals. In 29.4% of the SCC gastroscopies, no pathology could be identified.

Table 1 shows the patient characteristics, separated by outcome (cancer). Overall, the characteristics were similar across outcomes. Male sex was found to be more prevalent in the cancer group (*p* = 0.020). Furthermore, the average waiting time for evaluation by gastroscopy was slightly lower for patients who were subsequently diagnosed with cancer (*p* = 0.005).

### 3.2. Performance of SCC-EGC Criteria

With respect to the SCC-EGC criteria, dysphagia was not found to be associated with a statistically significant increased odds of upper GI cancer (OR 0.81; 95% CI 0.45–1.44), see Table 2. Even when narrowing down the definition to esophageal dysphagia, based on referrals that specified its location, there was still no statistically significant association (OR 0.86; *p* = 0.649). From the 348 referrals which reported dysphagia, 233 (67.0%) specified its location as esophageal and 20 (5.7%) as oropharyngeal. The location of dysphagia could not be discerned from the referral text in the remaining cases. Dysphagia also had a low PPV of only 5.5% (5.6% for esophageal dysphagia) for cancer. No EGC cases were identified in patients with oropharyngeal dysphagia.

Of all seven SCC criteria for EGC, only three were associated with statistically significant increased odds for cancer (*p* < 0.05); early satiety, unintentional weight loss, and radiological findings suggestive of upper GI cancer, with ORs of 2.49 (95% CI 1.16–5.34), 1.93 (95% CI 1.09–3.33) and 7.90 (95% CI 4.36–14.34), respectively. Overall, the positive predictive values of all signs and symptoms were low, with radiological findings being the only one with a PPV exceeding 20% (24.5%). Gastrointestinal bleeding yielded a significantly lower OR in the cancer group (OR 0.33, 95% CI 0.13–0.85, *p* = 0.015.)

### 3.3. Referrals to the Department of Surgery

A total of 247 patients were referred to the department of surgery because of suspected upper GI cancer, see Figure 2a,b. Of these, 105 patients were excluded from analysis, most commonly because subsequent clinical and histopathological records did not support a cancer diagnosis or because an alternative cancer diagnosis was made. Verified upper GI cancers (*n* = 139) were subsequently categorized as SCC (*n* = 40) or non-SCC (*n* = 99), depending on whether the primary investigation was based on an SCC referral. Out of the 40 SCC investigations, 37 were SCC gastroscopies, and the remaining three were computed tomography due to serious unspecific symptoms. Non-SCC investigations were either performed acutely (*n* = 30; 30.3%) or non-acutely (*n* = 69; 69.7%).

### 3.4. Waiting Times and Characteristics in SCC and Non-SCC Groups

Patient characteristics were similar for both groups (Table 3). Symptom duration prior to seeking healthcare (patient delay) was similar between the groups, although data could not be ascertained in 17.5% and 25.3% of referrals (SCC and non-SCC, respectively). Diagnostic delay (interval from initial referral to cancer diagnosis) was approximately 14 days longer for patients evaluated via non-SCC pathways (*p* = 0.045).

Referral indications were relatively similar between the two groups, see Table 4. Early satiety and weight loss were found to be more prevalent in SCC referrals (*p* = 0.001 and *p* = 0.040, respectively). No other symptoms were found to be more prevalent in either group (*p* < 0.05). There was also no statistically significant difference in the prevalence of at least one specific upper GI alarm symptom (dysphagia, emesis, hematemesis, or early satiety), *p* = 0.484.

Primary tumor characteristics could not be determined in 12.5% and 6.1% of the SCC and non-SCC cases, respectively (Table 5). There was no statistically significant association between referral pathway and primary tumor characteristics. Metastasized cancer was slightly more prevalent in the non-SCC group (36.2% vs. 27.0%), although this difference was not statistically significant (*p* = 0.319). Metastasis could not be determined in 7.5% and 5.1% of SCC and non-SCC cases (Table 5).

## 4. Discussion

Fast-track pathways for urgent endoscopic evaluation in the presence of alarming upper GI symptoms have been adopted by several countries over recent years, including Sweden. They were implemented as an attempt to diagnose esophageal and gastric cancer earlier. The diagnostic efficacy of the fast-track pathway is unknown, as well as adherence to its guidelines. This study aimed to evaluate key aspects of the SCC-EGC fast track, including cancer yield in SCC referrals, the predictive value of the SCC-EGC entry criteria, and its impact on diagnostic delay and overall prognosis.

We found an EGC yield of 6.2% in patients referred to gastroscopy, according to the SCC-EGC. Overall, most SCC gastroscopies showed benign findings or, in nearly a third of cases, normal findings despite meeting one or several SCC criteria. The relatively low cancer yield is comparable with that of studies assessing “open-access” endoscopy (~5–10%) [24,25], although differences exist with respect to both entry criteria as well as the clinical context in which the fast track is applied. For example, these studies neither assessed referrals from secondary care nor evaluated certain alarming signs or symptoms, such as early satiety or radiological findings. One further distinction is that in RÖC, SCC-EGC referrals are reviewed by a gastroenterologist and can be rejected if there is a low suspicion of cancer. Thus, with this additional filter, as well as additional criteria with relatively high PPVs. (early satiety and radiological findings), one might expect a higher cancer yield in the Swedish model. However, this is not the case. A possible explanation might include the overall low cancer prevalence relative to alarm symptoms in the population.

Dysphagia, a cornerstone alarm symptom in the SCC-EGC, was not found to be associated with a statistically significant increased odds of upper GI cancer. In addition, it had a surprisingly low PPV (5.5%). This PPV is similar to what has been reported in other studies [26,27] and presumably reflects that dysphagia is a relatively common symptom, whereas upper GI cancer is far less prevalent in the population. Indeed, several benign conditions are also associated with dysphagia, such as esophagitis, infections, or motility disorders [28]. Furthermore, dysphagia is a relatively unspecific term and is associated with both esophageal and oropharyngeal conditions [29]. As seen in our study, physicians often omit clear definitions of dysphagia in referrals, which further limits the use of this symptom as an SCC criterion.

Surprisingly, GI bleeding was associated with a statistically significant decreased odds of upper GI cancer. This reflects that benign upper GI disorders, or lower GI disorders, can be a cause of bleeding as well. Even so, 9.4% of the malignancies presented with some form of GI bleeding. Overall, most current SCC criteria appear to have a limited positive predictive value for EGC. This inevitably results in a significant proportion of benign or normal findings, thus conferring an increased strain on healthcare resources.

Our findings suggest that only a few of the current SCC criteria are predictors of malignancy, primarily early satiety, radiological findings, and unintentional weight loss, although the latter is both an unspecific symptom and poorly defined in the guidelines, not to mention subject to recall bias. Our PPVs. of alarm symptoms were similar to those reported in a meta-analysis, although that study did not include early satiety [30].

It is uncertain whether any symptom not currently part of the SCC-EGC guidelines may be a useful predictor of malignancy; however, several other studies have demonstrated either a limited predictive value or sensitivity of alarm symptoms in general with respect to EGC [26,31,32]. Other common symptoms found in the referrals from our study, such as abdominal pain, heartburn, and altered bowel habits, were not associated with increased odds of EGC. However, such non-alarming symptoms might have been underreported by referring physicians, not because they were absent but because they were considered superfluous or insignificant. Thus, this material probably provides limited insight into the predictive values of other symptoms.

Among known risk factors for upper GI cancer [33,34], only male sex was associated with increased odds for cancer, while previous smoking was borderline statistically significant. There are probably several reasons why our results differed from those of other studies. Primarily, in cases where no risk factors could be identified in the data collection process, we assumed that they were absent. Furthermore, we simplified the definitions of some risk factors, e.g., smoking was never quantified, and alcohol abuse could seldom be verified. In addition, some of the data in the referral texts or medical records might have been either old or inaccurate.

Another key finding in this study was that cancer diagnosis through other diagnostic pathways was more common than via the SCC. Only 37 out of 139 (26.6%) patients were initially referred for an SCC gastroscopy, or 40 (28.8%) if all patients diagnosed via an SCC pathway were included. Interestingly, the prevalence of specific symptoms was largely similar in the two groups. SCC referrals were more often associated with more specific upper GI symptoms (dysphagia, emesis, early satiety), although this association was not statistically significant except for early satiety. In contrast, symptoms that also could originate from many other sites in the GI tract (abdominal pain/dyspepsia, anemia, GI bleeding) were more prevalent in the non-SCC group, although there was no statistically significant difference. Weight loss, an unspecific but alarming symptom, was more often present in the SCC group, probably because physicians are largely aware of this potentially ominous symptom and thus refer these patients for urgent evaluation.

Non-acute investigations comprised most non-SCC investigations. This subgroup also had a similar prevalence of alarm symptoms when compared to the SCC group. A likely explanation is that many referring physicians are not aware of current national guidelines.

Interestingly, although the diagnostic interval differed significantly between patients referred through the SCC and those via other pathways, there were no significant differences in terms of outcome with respect to the primary tumor stage or metastasis upon presentation. This finding is similar to what several other studies have reported, including the report from the Swedish Agency for Health Technology Assessment and Assessment of Social Services [12]. One possible explanation for this might be the importance of patient delay, as well as the fact that the disease often remains asymptomatic until advanced stages [16]. Other authors have expressed concern that because alarm symptoms occur late in the disease, they may not be suitable for the identification of early cancer [16,35].

From our results, one might infer that adherence to national guidelines is inadequate. Many physicians referred patients with alarm symptoms for inappropriate primary investigations, such as barium swallow. This introduces a significant diagnostic delay, not to mention unnecessary costs and patient suffering due to additional diagnostic procedures. It may, therefore, be appropriate to improve awareness of current SCC-EGC guidelines in the health care system [36].

One major strength of this study was its inclusion of both all SCC-EGC referrals in RÖC, as well as all EGC cases. This broad approach through two research questions yielded valuable insight into the SCC-EGC with respect to both its predictive role in upper GI cancer as well as its use and usefulness in clinical practice.

This study has a few limitations. Due to its reliance on medical records, which may be either incomplete or ambiguous, there is a risk of misclassification bias. Such errors may have been introduced in the process of data collection due to misinterpretation of records. However, this bias is likely to be equal in cancer and non-cancer patients. The study population was relatively small, and only patients from RÖC were included in the statistical analysis. Because local guidelines or practices may differ in other parts of Sweden, our results may not necessarily reflect the rest of Sweden.

As a gastroenterologist initially assesses referrals, it is possible that some were not prioritized for urgent evaluation or possibly even rejected based on low suspicion of cancer. These referrals would not be found in the registry containing SCC referrals. Therefore, it is possible that some referrals, potentially including SCC criteria, were not analyzed. The inclusion of these referrals would likely have further lowered the PPVs of alarm symptoms based on the low clinical suspicion of malignancy.

This study can yield a better understanding of diagnostic pathways for EGC in clinical practice, and by revealing shortcomings with current guidelines, studies like this may lead to an improvement in the fast-track referral pathways. Our findings suggest that early satiety, unintentional weight loss, and suspicious radiological findings should be included in fast-track pathways, whereas dysphagia, for example, should be better defined.

## 5. Conclusions

The current fast-track program for esophageal and gastric cancer in Sweden appears to result in a limited cancer yield. Most of the SCC criteria are poor predictors of upper GI cancer, except for abnormal radiology and early satiety. The term dysphagia is inadequate and should be defined in more detail. Furthermore, most of these cancers were identified via other diagnostic pathways, resulting in significant delays for patients waiting for evaluation. In addition, the SCC-EGC does not seem to have a positive effect on disease outcomes. A larger, nationwide study may be of value to confirm these findings.

## Figures and Tables

**Figure 1 diagnostics-13-03577-f001:**
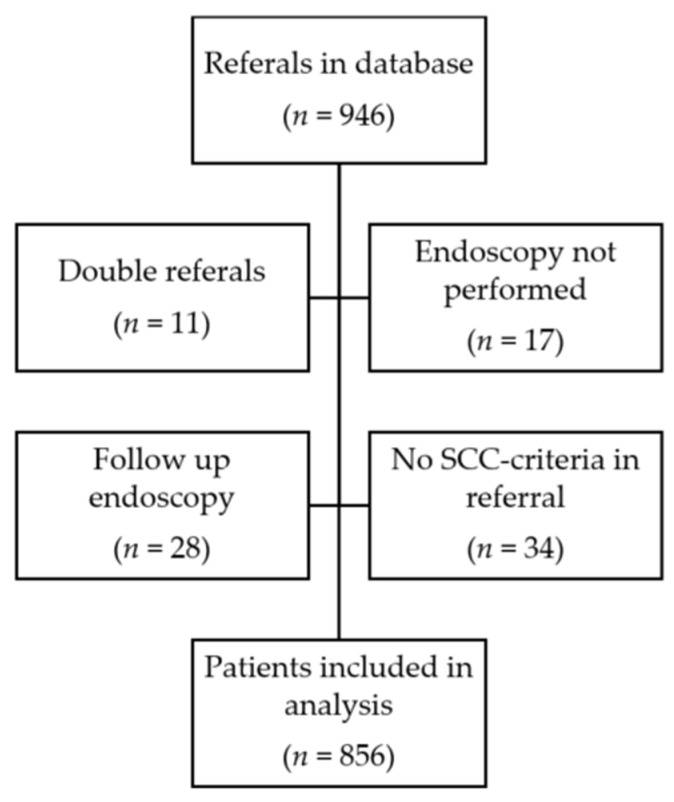
Flowchart of data collection and patient inclusion.

**Figure 2 diagnostics-13-03577-f002:**
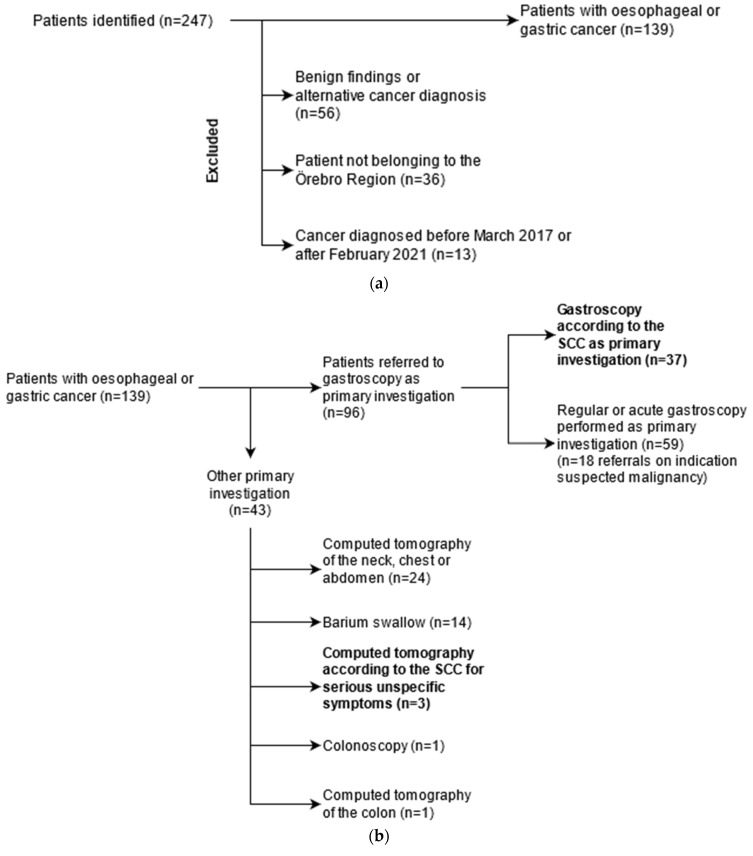
(**a**). Flowchart of data collection for all patients evaluated for oesophageal and gastric cancer. A total of 247 patients were identified in the Department of Surgery’s registry of oesophageal and gastric cancer cases in the RÖC from 2017. Patients who, after further investigation, were found to not have cancer (*n* = 56) were excluded from analysis. A number of patients (*n* = 36) belonged to another county than Örebro and were excluded. (**b**). Flowchart illustrating the primary (first) investigation for initial symptoms in oesophageal and gastric cancer patients. In total, 37 patients were diagnosed with gastroscopy according to the standardised course of care (SCC) guidelines. The majority of gastroscopies (*n* = 59) were performed either acutely or as a low-priority investigation. A minority of patients underwent other investigations, mainly computed tomography (*n* = 24) or barium swallow (*n* = 14). Three patients were evaluated for cancer according to the SCC for serious unspecific symptoms and were therefore included in the SCC-group.

**Table 1 diagnostics-13-03577-t001:** Baseline characteristics in patients evaluated by SCC gastroscopy.

Characteristic	No Cancer (*n* = 803)	Cancer (*n* = 53)	*p*-Value
Age, mean (SD)	67.8 (13.6)	71.1 (11.6)	0.145 ^
Waiting time *, days (range)	12.3 (0–308)	8.9 (1–54)	0.005 ^
Sex—male, *n* (%)	368 (45.8)	33 (62.3)	0.020
Smoking status			
Current, *n* (%)	89 (11.1)	8 (15.1)	0.372
Past, *n* (%)	105 (13.1)	12 (22.1)	0.050
Alcohol			
Current, *n* (%)	29 (3.6)	3 (5.7)	0.442
Past, *n* (%)	9 (1.1)	1 (1.9)	0.474
BMI ≥ 30, *n* (%)	173 (21.5)	13 (24.5)	0.610

All binary variables were analyzed with the Pearson Chi^2^-test or Fisher’s exact test if any parameter had an expected frequency < 5. ^ Continuous variables were analyzed using the Mann–Whitney U-test. * Waiting time from referral to evaluation by gastroscopy.

**Table 2 diagnostics-13-03577-t002:** Cancer frequency, positive predictive value (PPV), and Odds Ratio (OR) within a 95% Confidence Interval (CI) by gender, age, and SCC criteria in the referral.

Factors in Referral	No Cancer (*n* = 803) (%)	Cancer (*n* = 53) (%)	PPV, %	OR (95% CI)	*p*-Value
Sex—male	368 (45.8)	33 (62.3)	8.2	1.95 (1.10–3.46)	0.020
Age ≥ 50	723 (90.0)	51 (96.2)	6.6	2.82 (0.67–11.81)	0.138
SCC criteria					
Recent-onset dysphagia	329 (41.0)	19 (35.8)	5.5	0.81 (0.45–1.44)	0.462
(Esophageal dysphagia)	220 (27.4)	13 (24.5)	5.6	0.86 (0.45–1.64)	0.649
Emesis ^a^	119 (14.8)	10 (18.9)	7.8	1.34 (0.65–2.73)	0.425
Early satiety ^a^	61 (7.6)	9 (17.0)	12.9	2.49 (1.16–5.34)	0.016
Unintentional weight loss ^a^	283 (35.2)	27 (50.9)	8.7	1.91 (1.09–3.33)	0.021
Gastrointestinal bleeding (any)	192 (23.9)	5 (9.4)	2.5	0.33 (0.13–0.85)	0.015
Hematemesis	15 (1.9)	1 (1.9)	6.3	1.01 (0.13–7.80)	0.644
Iron deficiency anemia	220 (27.4)	8 (15.1)	3.5	0.47 (0.22–1.02)	0.050
Radiological findings (esophageal/gastric)	71 (8.8)	23 (43.4)	24.5	7.90 (4.36–14.34)	<0.001
(Any radiological finding suggestive of malignancy)	96 (12.0)	24 (45.3)	20.0	6.10 (3.41–10.90)	<0.001

^a^ Modified criteria. See data collection for a full explanation.

**Table 3 diagnostics-13-03577-t003:** Baseline characteristics of esophageal and gastric cancer patients referred via the standardized course of care (SCC) and those referred via other pathways.

Patient Characteristics	SCC (*n* = 40)	Non-SCC (*n* = 99)	Missing Cases (%) SCC/Non-SCC	*p*-Value
Male, *n* (%)	29 (71.2)	63 (63.6)		0.317
Age (years), mean (SD)	71.5 (11.8)	71.2 (10.6)		0.864 ^^
Risk factors				
Smoker, *n* (%)	4 (10.0)	18 (18.2)		0.232
Former smoker, *n* (%)	11 (27.5)	21 (21.2)		0.425
Obesity, *n* (%)	6 (15.0)	28 (28.3)		0.099
Primary care origin of referral, *n* (%)	30 (75.0)	52 (52.5)		0.015
Symptom duration (patient delay)			17.5/25.3	
<1 month	13 (38.2)	30 (40.5)		0.820
1–6 months	17 (50.0)	34 (45.9)		0.695
>6 months	3 (8.8)	10 (13.5)		0.751
Diagnostic interval, median (IQR) *	10 (10)	16 (36)		0.045 ^
≥1 SCC criteria in referral	38 (95.0)	90 (88.9)		0.347

Results are presented after excluding missing data from the analysis. * Defined as the interval from initial referral to cancer diagnosis. ^ Calculated with Mann–Whitney U-test; ^^ Calculated with unpaired *t*-test. All binary variables were analyzed with the Pearson Chi^2^-test or Fisher’s exact test if any parameter had an expected frequency < 5. Data distribution of continuous variables was determined using the Shapiro–Wilk test. SD—Standard Deviation; IQR—Interquartile Range.

**Table 4 diagnostics-13-03577-t004:** Symptoms in SCC and non-SCC referrals, respectively. The table lists all SCC criteria except for radiological findings.

Signs and Symptoms in Referral	SCC	Non-SCC	*p*-Value
Dysphagia	19 (51.4)	45 (44.1)	0.450
Emesis	9 (24.3)	17 (16.7)	0.306
Weight loss	21 (56.8)	40 (39.2)	0.065
Early satiety	7 (18.9)	3 (2.9)	0.004
GI bleeding	4 (10.8)	17 (16.7)	0.394
Anemia	7 (18.9)	27 (26.5)	0.360
Abdominal pain/dyspepsia	10 (27.0)	45 (44.1)	0.069
Reflux	7 (18.9)	9 (8.8)	0.131
Altered bowel habits	1 (2.7)	6 (5.9)	0.675
≥1 Specific alarming upper GI symptom *	24 (64.9)	60 (58.8)	0.520

All binary variables were analyzed with the Pearson Chi^2^ test or Fisher’s exact test if any parameter had an expected frequency < 5. * Dysphagia, emesis, hematemesis, or early satiety.

**Table 5 diagnostics-13-03577-t005:** Features of malignancies identified through SCC and non-SCC pathways, respectively. Classification according to the TNM system.

Cancer Characteristics	SCC (*n* = 40)	Non-SCC (*n* = 99)	Missing Cases (%) SCC/Non-SCC	*p*-Value
Primary tumor			12.5/6.1	
T1	3 (8.6)	2 (2.2)		0.125
T2	13 (37.1)	31 (33.3)		0.686
T3	13 (37.1)	39 (41.9)		0.623
T4	6 (17.1)	20 (21.5)		0.585
Metastasis	10 (27.0)	34 (36.2)	7.5/5.1	0.319

Results are presented after excluding missing data from the analysis. All binary variables were analyzed with the Pearson Chi^2^ test or Fisher’s exact test if any parameter had an expected frequency < 5.

## Data Availability

The data presented in this study are available on request from the corresponding author.

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
