# Peer review of "The Swedish Standardized Course of Care—Diagnostic Efficacy in Esophageal and Gastric Cancer"

_diagnostics, 2023, doi:10.3390/diagnostics13233577_

Round 1

Reviewer 1 Report

Comments and Suggestions for Authors

Thank to the Editors to review this paper. This is a very nicely written paper about the diagnostic efficacy of gastric and esophageal cancer. I think it is worth publishing in Diagnostics, but some minor points are missing

Abstract: If you divide the abstract into parts, you should follow it throughout. I can see only "results".
The authors should expand all the abbreviations if they are used for the first time

Introduction: OK

Material and Methods: Who performed gastrocopy? Which TNM was used?

Results: Please add the number of patients in each group to the Table 1

Discussion, results: ok

Author Response

Thank you very much for taking the time to review this manuscript and your positive opinion. Please find the detailed responses below and the corresponding revisions in the re-submitted manuscript.

Abstract: If you divide the abstract into parts, you should follow it throughout. I can see only "results".
The authors should expand all the abbreviations if they are used for the first time

Abstract has been corrected so that there are no headings. The Abbreviation positive predictive value (PPV) has been expanded.

Material and Methods: Who performed gastrocopy? Which TNM was used?

We have clarified who performed the gastroscopies by adding a sentence regarding this.

We have specified which version of the TNM system used.

Results: Please add the number of patients in each group to the Table 1

Number of patients in each group was added to table 1.

Reviewer 2 Report

Comments and Suggestions for Authors

The authors compare the SCC and the fast-track referance to diagnose EGC. It is a way for both early cancer diagnosis and unload stresses in the health system all over the countries.  It is written in good shape and answers many questions. I just have a few comments:

Statistical analysis: Sample size should be calculated.

Add a paragraph to mention your suggestions to improve the diagnostic yield for both SCC and fast-track ways.

Is there any data comparing the real costs of either diagnostic way; and which one is less costly?

Author Response

Thank you very much for taking the time to review this manuscript and your positive opinion. Please find the detailed responses below and the corresponding revisions in the re-submitted manuscript.

Statistical analysis: Sample size should be calculated.

In this study, all patients referred for gastroscopy according to SCC-EGC criteria and all known EGC-cases in our County was included. Therefore, calculating a sample size was not feasable.

Add a paragraph to mention your suggestions to improve the diagnostic yield for both SCC and fast-track ways.

We have added a paragraph at the end of the discussion according to your suggestions.

Is there any data comparing the real costs of either diagnostic way; and which one is less costly?

The health care burden of performing gastroscopies and other investigations to exclude gastrointestinal malignancies are considerable, with a large number of patients needed to scope to diagnose a malignancy. However, we do not have any data on the actual costs or if there are differences in diagnostic pathways.